# Kinetic Processes in Enzymatic Nanoreactors for In Vivo Detoxification

**DOI:** 10.3390/biomedicines10040784

**Published:** 2022-03-27

**Authors:** Zukhra Shajhutdinova, Tatiana Pashirova, Patrick Masson

**Affiliations:** 1Biochemical Neuropharmacology Laboratory, Kazan Federal University, Kremlevskaya Str. 18, 420111 Kazan, Russia; shajhutdinova.z@mail.ru; 2Arbuzov Institute of Organic and Physical Chemistry, FRC Kazan Scientific Center, Russian Academy of Sciences, Arbuzov Str. 8, 420088 Kazan, Russia; tatyana_pashirova@mail.ru

**Keywords:** bioscavenger, confined-reaction space, detoxification, enzyme, nanoreactor, organophosphate, second-order reaction

## Abstract

Enzymatic nanoreactors are enzyme-encapsulated nanobodies that are capable of performing biosynthetic or catabolic reactions. For this paper, we focused on therapeutic enzyme nanoreactors for the neutralization of toxicants, paying special attention to the inactivation of organophosphorus compounds (OP). Therapeutic enzymes that are capable of detoxifying OPs are known as bioscavengers. The encapsulation of injectable bioscavengers by nanoparticles was first used to prevent fast clearance and the immune response to heterologous enzymes. The aim of enzyme nanoreactors is also to provide a high concentration of the reactive enzyme in stable nanocontainers. Under these conditions, the detoxification reaction takes place inside the compartment, where the enzyme concentration is much higher than in the toxicant diffusing across the nanoreactor membrane. Thus, the determination of the concentration of the encapsulated enzyme is an important issue in nanoreactor biotechnology. The implications of second-order reaction conditions, the nanoreactor’s permeability in terms of substrates, and the reaction products and their possible osmotic, viscosity, and crowding effects are also examined.

## 1. Introduction

For some 50 years, enzyme-supported systems have been used in analytical chemistry, clinical biology, and environmental and forensic toxicology where immobilized enzymes act as sensors. Biosensor technology has been utilized to detect and monitor the presence of specific chemicals in biological fluids and the environment. However, the nanoreactor concept is more recent, and applications of nanoreactor (nR) technology are currently emerging. In nanoreactors, one or several encapsulated enzymes is/are capable of performing either a single reaction (synthesis or degradation), coupled reactions, or consecutive reactions. In the most complex systems, nanoreactors may contain multiple components (enzymes, substrates, cofactors) acting in reaction chains, as in metabolic processes. As an analogy, mitochondria, the energy power plant of the cell, which is capable of metabolizing the end-products of glycolysis and fatty acids and producing energy, can be considered to be complex natural nanoreactors. Likewise, artificial therapeutic nanoreactors are becoming more and more complex, the ultimate goal being to create artificial organelles [1], cells, and tissues for correcting genetic defects, treating metabolic and degenerative diseases, and repairing physiological functions and organs.

Enzyme-catalyzed reactions in therapeutic nanoreactors that operate in the bloodstream have been investigated for years and have already been used in medical and pharmacological applications, yet the formal mechanisms of enzyme reactions in such nanobodies have been little documented [2]. However, numerous advances have already been made regarding the neutralization of endogenous toxicants, e.g., ROS [3], uric acid [4], exogenous toxicants like drugs [5], and industrial poisons such as cyanide [6]. In principle, all kinds of xenobiotics and pollutants can be enzymatically degraded. The main enzyme nanoreactors of medical interest for the detoxification of endogenous and exogenous toxicants have recently been reviewed in the literature [3,4,5,6,7]. The neutralization of toxic compounds, (T), involves either stoichiometric, catalytic, or pseudo-catalytic processes.

Stoichiometric, pseudo-catalytic, and catalytic detoxification processes have recently been reviewed, with special emphasis on the inactivation of organophosphorus compounds (OP) [7]. The concentration of toxic substrates in natural media and blood is in general very low, much lower than the enzyme-substrate dissociation complex constant, while the concentration of encapsulated enzymes is high. Moreover, the diffusion of substrates across the nanoreactor membranes is diffusion-controlled. Therefore, under such conditions, neutralization reactions—those within the confined volume of nanoreactors—take place under second-order conditions. In the present work, we discuss the basis, practical consequences, and applications of such reaction processes in confined spaces.

## 2. Principles of Detoxification Processes by Exogenous Enzymes: Stoichiometric Neutralization versus Catalytic and Pseudo-Catalytic Degradation of Toxicants

Depending on the origin of the toxicant (exogenous or endogenous) and the route of poisoning (inhalation, ingestion, injection, skin penetration, or the cellular mechanisms of production), the concentration of the toxicant in the blood compartment, [*T*], is controlled by diffusion from peripheral compartments, the reaction with endogenous bioscavengers, and physiological targets and elimination. [*T*], as a function of time, displays skewed distributions. Even in the most severe case of poisoning, the maximum concentration of toxicants in the blood is very low. For example, in the case of poisoning by OPs, in most severe cases, the maximum toxicant concentration never exceeds the µM range (e.g., < 500 nM for paraoxon, the active metabolite of the pesticide parathion) [8] (Figure 1).

At this point, we must consider two cases for the enzymatic degradation of toxicant molecules: (a) the inactivating enzyme is freely circulating in the bloodstream and, thus, the detoxification process takes place in plasma; (b) the enzyme is encapsulated in a stable nanoreactor, circulating in the bloodstream, and the detoxification process occurs inside the nanoreactor chamber. The principles of volume-confined enzyme-catalyzed reactions, such as in nanoreactors, have been thoroughly described elsewhere [9]. In addition, a few theories about kinetic processes in enzymatic nanoreactors have been recently summarized [2].

Here, we will elucidate the formal principles that control the kinetics of toxicant inactivation.

### 2.1. In Vivo Detoxification by Free Enzymes

Enzyme therapy has been thoroughly investigated for decades, in particular for its implementation as an antidotal therapy in the case of drug overdose or acute poisoning by xenobiotics [5]. Detoxification can result either from stoichiometric neutralization, catalytic inactivation, or pseudo-catalytic inactivation (Figure 1). In the first case, the mole-to-mole reaction between the enzyme (E) and the toxicant (T) is accompanied by the release of the leaving group, X (for example, in OP structures (presented in Figure 1), the X of paraoxon is para-nitrophenol, the X of echothiophate is thiocholine, and the X of sarin is a fluoride ion). It leads to an enzyme conjugate (E-T’); the enzyme is irreversibly inhibited (Figure 1a). This implies the use of large amounts of bioscavengers for complete neutralization. On the other hand, in catalytic inactivation processes, the toxicant acts as a substrate and the reaction occurs with a turnover until completion; the active enzyme is recycled after each turnover; therefore, much less enzyme is needed than in stoichiometric neutralization. In pseudo-catalytic inactivation, the covalent enzyme conjugate is reactivated in a coupled reaction.

To simplify, formal mechanisms are assumed to be Michaelian. For the three mechanisms, the first step is the formation of the reversible enzyme-toxicant complex (*E.T*). The dissociation constant, *K_D_*, is the ratio between the rate of dissociation (*k_off_* or *k*_−__1_) of *E.T* and the rate of its formation (*k_on_* or *k*_1_): *K_D_* = *k*_−__1_*/k*_1_.

#### 2.1.1. Catalytic Neutralization

First, let us consider the catalytic detoxification mechanism. Assuming that in vivo enzymatic degradation of toxic substrates (*T*) obeys the simple Michaelis–Menten kinetic model, leading to the release of the non-toxic product of T, X, and T’ (Figure 1c), the velocity (*v*) is established with Equation (1):(1)$$v=\frac{k_{\mathrm{cat}}\left[E\right]\left[T\right]}{K_{m}+\left[S\right]}.$$

In the minimum mechanistic scheme (Figure 1c), assuming a rapid equilibrium between the free enzyme and substrate *T*, i.e., *k*_2_ ≪ *k*_−__1_, the catalytic parameters are:(2)$$k_{\mathrm{cat}}=\frac{V_{max}}{\left[E\right]},\;\mathrm{and}\;K_{m}=\frac{\left(k_{-1}+k_{2}\right)}{k_{1}}=K_{D}+\frac{k_{2}}{k_{1}}$$

When there is an intermediate kinetic step after the formation of the *ES* complex (*E.T′*), this intermediate is subsequently hydrolyzed by water, acting as a co-substrate (Figure 1d). This mechanism is used by carboxylesterases and cholinesterases that hydrolyze toxic esters, such as cocaine, heroin, carbamates, and organophosphates. The catalytic parameters for this mechanism are:(3)$$k_{\mathrm{cat}}=\frac{V_{max}}{\left[E\right]}=\frac{k_{2}k_{3}}{k_{2}+k_{3}},\;\mathrm{and}\;K_{m}=\frac{\left(k_{-1}+k_{2}\right)k_{3}}{k_{1}\left(k_{2}+k_{3}\right)}=\frac{K_{D}k_{3}}{\left(k_{2}+k_{3}\right)}$$

For both catalytic mechanisms:*k_cat_/K_m_ = k*_2_*/K_D_*(4)

In mechanisms 1c and 1d, *k_cat_* is the catalytic turnover, *K_m_* is the Michaelis constant, and [*E*] is the actual enzyme concentration in the blood. Under these conditions, *k_cat_*/*K_m_*, the specificity constant is the bimolecular rate constant (second-order rate constant, M^−1^min^−1^) of the reaction. At very low concentrations of toxicant, where [T] ≪ *K_m_*, the velocity is:(5)$$v=\frac{k_{\mathrm{cat}}}{K_{m}}\left[E\right]\left[T\right]$$

The product (*k_cat_/K_m_*). [*E*] can be regarded as the first-order rate constant (min^−1^) of the reaction. Then, the concentration of the toxic substrate,$\;{\left[T\right]}_{t}$, at any one time is:(6)$$[T{]}_{t}=[T{]}_{0}\exp\left(-\frac{k_{\mathrm{cat}}{\left[E\right]}_{0}}{K_{m}}\cdot t\right)$$

Therefore, the time (*t*) needed to drop the toxicant concentration level below the toxic concentration threshold depends on the concentration of the degrading enzyme, ${\left[E\right]}_{0}$, and its bimolecular rate constant *k_II_* = *k_cat_/K_m_.*

#### 2.1.2. Stoichiometric Neutralization

In the case of stoichiometric neutralization by reactants (*R*), e.g., antibodies, specific reactive proteins, target enzymes, and chemically reactive traps such as functionalized cyclodextrins (Figure 1a), the first-order rate of reaction also depends on both the bimolecular rate constant (*k_II_ = k_i_ = k*_2_*/K_D_*) and reactant concentration [*R*]. Then, catalytic and stoichiometric detoxifications are first-order in *T*: [*E*] and [*R*] ≪ [*T*]. Comparing stoichiometric neutralization with catalytic degradation of identical efficacy, i.e., processes that display the same rate of detoxification, −*d*[*T*]/*dt*, it follows that:*k_i_*[*R*]_0_ = (*k_cat_*/*K_m_*)[*E*]_0_(7)

Owing to the high cost of biopharmaceuticals, only small doses of exogenous enzymes or stoichiometric reacting proteins can be administered for detoxification under affordable and safe medical circumstances. Therefore, the bimolecular rate constant *k_i_* of stoichiometric bioscavengers, or the specificity constant *k_cat_/K_m_* of catalytically acting detoxifying enzymes (catalytic bioscavengers), must be as high as possible for them to act in the shortest time. Moreover, injected biopharmaceuticals must be stable in the bloodstream and must display slow pharmacokinetics of elimination so that their concentration is maintained to be high for as long as possible. Lastly, by virtue of turnover, enzyme-catalyzed detoxification is economically much more interesting than stoichiometric neutralization, so that if *k_cat_/K_m_ > k_i_*, then [*E*]_0_ *<* [*R*]_0_. Progress in the genetic engineering of proteins, computer-(re)-design, directed evolution, and the biotechnology of enzymes allows us to meet this challenge.

#### 2.1.3. Pseudo-Catalytic Neutralization

Pseudo-catalytic neutralization is a hybridization between stoichiometric and catalytic mechanisms. In this system, two consecutive reactions are coupled (Figure 1b). In the simplest consecutive reactions (Figure 2), the toxicant T is first inactivated (*T* → *R*_1_), and in a coupled reaction, the product *R*_1_ is converted to a second product, *R*_2_ (*R*_1_ → *R*_2_) with the concomitant regeneration of *E*:

In the case of first-order reactions:−*d*[*T*]/*dt* = −*d*[*E*]/*dt* = *k*_1_[*T*](8a)
*d*[*E*-*R*_1_]/*dt* = *k*_1_[*T*] − *k*_2_[*E-R*_1_](8b)
*d*[*R*_2_]/*dt* = *−k*_2_[*E-R*_1_]*R*_1_= *d*[*E*]/*dt*(8c)

In this simple, irreversible consecutive reaction system, the concentration of intermediate *E-R*_1_ increases first, but if the rate constant of the coupled reaction *k*_2_ = *k*_1,_ then *d*[*E-R*_1_]*/dt* = 0, and [*E*-*R*_1_] remains small and roughly constant during the steady-state period. If *k*_2_
*< k*_1_, *k*_2_ is rate-limiting and *E-R*_1_ transiently accumulates. If *k*_2_ > *k*_1_, *E-R*_1_ does not accumulate. In the case of cyclic consecutive reactions where the production of *R*_2_ is accompanied by regenerated *E*, if *k*_2_ ≫ *k*_1_, there is a pseudo-turnover.

Therefore, to convert an *OP* stoichiometric bioscavenger (*E*) to a pseudo-catalytic bioscavenger, after the neutralization of *OP* by E with the release of the leaving group X- and *E-OP’*, the phosphylated enzyme (i.e., *E*-*R*_1_, in Figure 2), the phosphylated enzyme has to be reactivated in a coupled reaction, using a reactivator (*R*, a nucleophile compound). The nucleophilic attack of *R-O*: displaces the phosphyl moiety from the enzyme active center, leading to regeneration of the free active enzyme (Figure 2).

The pseudo-turnover depends on the efficacy with which the reactivator reactivates the phosphylated enzyme. The reactivation efficiency is controlled by the reactivator concentration ([*R*]), with [*R*] ≫ [*E*], the dissociation constant (*K_D_*) of phosphylated enzyme reactivator complex, and the kinetic parameters of the reaction (reactivation rate constant, *k_r_*, and bimolecular rate constant of reactivation, *k_r_/K_D_*) [10] (Figure 3).

Accordingly, the observed first-order rate of reactivation (*k_obs_*) is:(9)$$k_{\mathrm{obs}}=\frac{k_{r}\;\left[R\right]}{K_{D}+\left[R\right]}$$
with a half-time of reactivation:*t*_1/2_*= Ln*2*/k_obs_*(10)

However, it should be noted that if the phosphorylated enzyme concentration is high, close to that of the reactivator, [*R*] ≈ [*E-OP’*], the reactivation process will take place under second-order conditions. Such a situation may occur if a high concentration of toxicant *T* is neutralized in a short time by the enzyme. Reactions inside nanocontainers containing both enzyme and reactivator may meet this constraint. However, in the case of butyrylcholinesterase (BChE) as an OP bioscavenger, until recently, the rates of reactivation of the phosphylated enzyme by oximes were too slow to fulfill this condition. Research on new oximes that allow the faster reactivation of BChE is of paramount interest for generating BChE-based pseudo-catalytic bioscavengers [11,12].

Another limitation to the use of ChE is that with certain OPs, a post-phosphylation reaction may occur, leading to the non-reactivatability of phosphylated ChEs. This reaction, called “aging”, results from a partial dealkylation of the OP adduct [13]. To overcome this complication, mutants of ChEs that do not age or that display a very slow aging rate may be used [14].

### 2.2. Detoxification by Enzymes Encapsulated in Circulating Nanoreactors

The encapsulation and capping of therapeutic (antidotal) enzymes against drugs, narcotics and toxicants have been introduced in medicine for many years with the aim of increasing the operational stability and residence time of administered enzymes and preventing the triggering of the immune response in heterologous organisms [5,6,15,16]. However, little attention has been paid to the physicochemical conditions of reactions or to the formal catalytic mechanisms of enzymes that operate in nanoreactors (nR) [9]. In most cases, it is not clear whether the reactions take place inside nanocontainers or in the free solution after the dislocation of the membrane structure. In our approach, it is assumed that the nR envelopes are stable and that the enzyme reactions take place in the nR body. Therefore, for enzyme-catalyzed detoxification processes occurring in the small, confined volume of a nanoreactor, toxic molecules, T, must first diffuse from the blood compartment to the nR body. Moreover, high concentrations of the enzyme(s) are present in this small volume. Thus, the detoxification reactions take place under second-order conditions ([*E*] ≫ [*T*]), controlled by the diffusion of T molecules across the nanoreactor envelope.

Assuming nR is a sphere of radius *r* (Figure 3), enzyme-catalyzed reactions inside this volume depend on substrate partition between the bulk solution (i.e., blood) and the nanoreactor core, the diffusion of both the substrate, *T*, and product(s), *P*, across the nanoreactor membrane, and enzyme concentrations inside the nanoreactor. All these physical constraints affect the catalytic parameters and may alter the enzyme’s catalytic mechanisms. An important issue in the design of nanoreactor polymeric membrane is, therefore, how to allow the free diffusion of substrates *in* and reaction products *out* of the nR compartment, to prevent production inhibition of the enzyme, either by excess substrate or by the accumulation of reaction products.

#### 2.2.1. Structure of Enzyme-Containing Nanoreactors and their Application to Improve Biocatalytic Reactions

As we mentioned in a recent review [7], the most well-known synthetic enzyme-containing nanoreactors are liposomes, polymeric vesicles (polymersomes, PICsomes, CAPsomes, dendrimersomes) [17,18,19,20,21], hybrid materials [22], metal–organic frameworks (MOFs) [23,24,25,26,27], enzyme–polymer complexes [28], nanotubes, and silica nanoreactors [29]. Natural structures such as bacteria, cells, and viruses have also been used as nanoreactors, but they are out of the scope of this paper. This section is devoted to important features for increasing the activity of biocatalytic nanoreactors, factors affecting reaction rates, and reaction mechanisms in nanoreactors:-Increasing enzyme-loading into the nanoreactor [30]: This can be achieved by: (i) the modification of enzymes for strengthening their binding in nanoreactor, which can be carried out by covalent bonds with small molecules or polymers, electrostatic bonds and their immobilization on polymer, and by using cross-linking agents. One of the most successful examples is of three enzymes covalently encapsulated into a substrate-permeable silica nanoreactor by a mild fluoride-catalyzed sol–gel process [29]. (ii) The multicompartment loading of enzymes and the sub-compartmentalization of several enzymes by encapsulation within the lumen, entrapment in the membrane, and conjugation on the surface may yield high efficiency. This can be achieved by the cross-linking of enzymes with the building blocks of nanoreactors, or by capturing them in a layer-by-layer assembly. Incompatible enzymes can be encapsulated in nanoreactor sub-compartments. In addition, the mutual templated crystallization between enzymes and the material of the nanoreactor prevents enzyme molecules from unfolding [31]. Recently, a dual-confinement strategy that can controllably confine a series of enzymes in a nanocage-based zeolite imidazole framework [32] and multi-shelled metal-organic frameworks [33] were developed. (iii) The control of nanoreactor self-assembly by the regulation of parameters—packing parameter (*p*) and polymer hydrophilicity (ƒw)—and using improved innovative methods for nanoreactor biotechnology. Thus, microfluidics presents a new and promising method, providing high encapsulation efficiency and scaling [34].-The permeability to reactants and reaction products needs to be improved. This can be achieved through different strategies: (i) the selective selection of nanoreactor building blocks. Several methods of quantifying membrane permeability (fluorescence spectroscopy, osmotic swelling, pulsed-field gradient NMR spectroscopy) and the passage of molecules have been proposed [35]. Usually, lipid vesicles are highly permeable. Further strategies are associated with the production of organo-inorganic hybrid and polymeric vesicles. Polymeric vesicles are generally impermeable, due to their increased thickness and low membrane lateral diffusion. Diffusion is primarily associated with different structural conformations of the macromolecules in the membrane. Thus, hindered diffusion was observed for copolymers of high molecular weight [36]. The permeability of hybrid unilamellar polymer/lipid vesicles is mainly determined by the lipid-polymer composition [37]. The influence of the molecular weight and architecture of block copolymers on the linear tension and its consequences for the membrane structuring of hybrid polymer-lipid vesicles are presented in the literature [38]. One innovative strategy is switching between the impermeable and a semipermeable state of the vesicle membrane. For this purpose, functional groups were introduced into the block copolymer membrane for increasing its polarity in response to mechanical action, thereby increasing the permeability of the membrane to water-soluble substrates [39]. Another new approach is the incorporation of the polymer into a phospholipid membrane, where it acts as a synthetic molecular channel. Imparting an anionic charge to the vesicle’s surface represents a simple and versatile approach to substrate sorting and enhances molecular permeability [40]. The unique lattice membrane of the vesicles was made up of hundreds of small polymeric vesicles, linked together through multiple interactions [41]. (ii) The application of stimulus-responsive materials for the synthesis of nanoreactors, such as temperature, light, solvents, pH, and light [18]. Such changes often alter permeability or the phase separation dynamics; thus, they can be used to effectively turn on or off enzymatic reactions. Examples of photo-cross-linked and pH-sensitive materials for polymersomes are presented in the literature [42]. The bidirectional feedback mechanism of chemoenzymatic pH clock-mediated transient assembly regulates the existence of the transient state and controls the activity of the assembly [43]. Stimulus-responsive linkers were integrated into the cross-linking membrane network of nanoreactors, based on polyion complex vesicles [44]. The permeability of polymer-functionalized epoxy membranes was modulated by opening the epoxy ring and various diamine crosslinkers and hydrophobic primary amines [45]. (iii) The insertion of membrane proteins and the formation of membrane protein channels [46] via peptides that form bio-pores. Thus, the design of “intelligent” membranes with precisely controlled and sensitive behavior may provide solutions to increase the efficiency of nanoreactors.

#### 2.2.2. Concentration of Enzymes inside the Nanoreactor

For the quantitative analysis of reactions taking place in *nR*s, the concentration of the encapsulated enzyme, [*E*] must be known. Several methods can be used to access [*E*]. The enzyme concentration inside each nanoparticle is estimated using the nanoparticle concentration (quantification of the number of particles per sample volume) in the solution. This concentration inside nanoreactors (*nR*) can be calculated using Equation (11):(11)$$\left[E_{\mathrm{nR}}\right]=\frac{V_{\mathrm{all}}\left[E_{\mathrm{all}}\right]}{V_{nR}}$$
where [*E_all_*] is the amount of enzyme loaded into the *nR* preparation, taking into account the enzyme encapsulation efficiency (EE, %); *V_nR_* is the volume occupied by the nanoreactors in the preparation, with *V_all_* as the total volume and *N_nR_* as the nanoreactors’ concentration (particle number concentrations/mL).

The calculation is based on the restrictive assumptions that the *nR* geometry and distribution are spherical, monodisperse, and monomodal. The volume *V_nR_* (Figure 4) occupied by all *nR* particles of diameter (*d*) can be calculated using Equation (12):(12)$$V_{\mathrm{nR}}=N_{\mathrm{nR}}\frac{4}{3}\pi {\left(\frac{d}{2}\right)}^{3}$$
where $\frac{4}{3}\pi {\left(\frac{d}{2}\right)}^{3}$ is the volume of spherical *nR*.

Thus, knowing the volume of the preparation, given the volume occupied by all *nR* particles in the sample (*V_nR_*), it is possible to calculate the enzyme concentration inside the nanoreactor [*E_nR_*].

A theoretical calculation of particle number concentrations was proposed by the authors of [47]. The nanoreactor concentration, *N_nR_*, can be expressed as particle number concentrations. However, at present, there are no certified standards for the determination of particle concentration or reference materials for the calibration of measuring concentration [48,49,50]. However, there are many empirical methods that are widely used for the determination of particle concentration, including optical and microscopy methods: in particular, atomic force microscopy [51], turbidimetry [52], counting methods, gravimetry [53], and methods based on light-scattering. Innovative method combinations are being developed, such as tunable resistive pulse sensing [54], nanoflow cytometry, multi-angle dynamic light scattering [55], small-angle X-ray scattering (SAXS) [56], centrifugal liquid sedimentation methods, flow fractionation online with a multi-angle light scattering detector (AF4-MALS), and a simple, non-destructive method suitable for the rapid evaluation of the number concentrations of composition-homogeneous, non-absorbing nanoparticles in colloidal suspension [57].

Electrospray (ES) is used to generate vapor-phase-dispersed material, followed by sizing and counting the number of nanoparticles using a scanning particle mobility meter (SMPS). Ion mobility can be analyzed via differential mobility analysis (DMA), gas-phase electrophoretic molecular analysis (GEMMA), or mass spectrometry (MS) methods [58,59]. Nano-electrospray gas-phase electrophoretic mobility molecular analysis (nES GEMMA) also presents an alternative method [60]. However, a comparative evaluation using several methods to ensure an accurate and reliable determination of particle concentration is recommended. Specifically, this approach was applied for determining the concentrations and size distributions of PEGylated liposomes, with NTA, TRPS, and AF4-MALS. The results were in good agreement [48].

However, the use of certain methods depends on the type of nanoparticles, and all have some limitations and drawbacks. For example, UV-visible spectrophotometry (UV Vis) is fairly accurate for determining the concentration of metallic nanoparticles [61]. For carbon nanodots, the best method is UV-Vis and fluorescence emission spectrophotometry [62]. Measurements with single-particle inductively coupled plasma mass spectrometry (spICP-MS) is applicable only for nanoparticles with element labels visible to ICP-MS within lower detection limits. For sub-micrometer-sized nanoparticles, the accuracy is limited [63] and it is better to combine this method with the novel dynamic mass flow approach [64]. Particle tracking analysis (PTA) is suitable for all nanoparticles but exhibits reduced sensitivity when encountering the lower detection limit of particle size. The presence of particle agglomeration complicates the measurement of concentration by PTA.

The reliable method to establish the concentration of encapsulated enzymes is nanoparticle tracking analysis. Nanoparticle tracking analysis (NTA), based on light scattering, tracks the Brownian motion of each particle individually to determine the root mean square (rms) displacement of individual particles [65,66]. Nanoparticle concentration is then determined by the number of tracked particles in the estimated volume of illumination [67].

#### 2.2.3. Kinetic Principles of Enzyme Reactions in Nanoreactors

The basic theory of entrapped-enzyme catalysis was established by Laidler and his coworkers in the early 1970s [68,69] and may be applied to the catalytic reactions inside nR.

The ratio between substrate concentrations in the bulk solution, [*T*], and at the nR interface, [*T’*], defines the partition coefficient, *P*. In the absence of the enzyme, the rate of accumulation of the toxicant substrate, due to diffusion across the nanoreactor envelope, obeys Fick’s second law (Equation (13)). The toxicant concentration in the nanoreactor progressively increases and then reaches [*T_nR_*], a non-reactional steady state after time *τ*. The lag time, *τ*, depends on the diffusion coefficient, *D*, of molecules T diffusing in the nanoreactor.
(13)$$\frac{\partial \left[T_{\mathrm{nR}}\right]}{\partial t}=D\frac{\partial^{2}\left[T_{\mathrm{nR}}\right]}{\partial r^{2}}$$

The diffusion coefficient, *D*, of molecules, *T*, is related to dynamic viscosity by the Stokes–Einstein equation (Equation (14)):(14)$$D=\frac{k_{B}T}{6\pi \eta r_{T}}$$
where *k_B_* is the Boltzmann constant, ***T*** is the absolute temperature (K), and *r_T_* is the molecular radius of the toxicant molecule (considered as a sphere).

Thus, [*T_nR_*] is the function of time and the *nR* radius, *r*, according to Equation (15) [70]:(15)$$\frac{\left[T_{\mathrm{nR}}\right]}{\left[T{}^{\prime }\right]}=1-\left(\frac{4}{\pi }\right)\sin\frac{\pi }{2}\exp\left(-\frac{D\times \pi^{2}\times \tau }{r^{2}/4}\right)$$

The non-reactional steady state is reached when [*T_nR_*] differs from [*T’*] by ± 10%, after a short lag time; *τ* = 0.257 (*r*^2^/4*D*), where *D* is the range of 10^−9^ m^2^·s^−1^, as for most small molecules in solution. In the presence of the encapsulated enzyme into *nR*, the substrate concentration, ([*T_nRE_*]), which is lower than [*T_nR_*], the non-reactional steady state will progressively reach a reactional steady state.

When the pseudo-steady state is reached, the substrate diffusion and enzymatic reaction become interdependent. The rate of substrate influx across the nanoreactor section is proportional to the concentration gradient *d*[*T_nR_*]/*dr* and is equal to the enzymatic rate (*v′*). Assuming that the enzyme obeys a simple Michaelis–Menten mechanism (the reactions are shown in Figure 1), Equation (16) describes the rate of reaction as a function of [*T_nR_*]:(16)$$D\frac{\partial^{2}\left[T_{\mathrm{nR}}\right]}{\partial r^{2}}=\frac{k{{}^{\prime }}_{\mathrm{cat}}\;\left[E\right]\left[T_{\mathrm{nR}}\right]}{K{{}^{\prime }}_{m}+\left[T_{\mathrm{nR}}\right]}$$

The apparent catalytic parameters, *k′_cat_* and *k′_m_*, differ from the catalytic parameters in terms of the reaction in free solution (*k_cat_* and *K_m_*; c.f. (Equations (1)–(5)) because of environmental constraints (nano-volume, crowding, etc.). In the case of the progressive inhibition of enzymes by pseudo-substrates, i.e., in terms of stoichiometric bioscavengers (Figure 1a), where the saturation is Michaelian, Equation (16) applies, with *k′_m_* being replaced by *k′_D_* and the catalytic bimolecular rate constant *k′_cat_/k′_m_* being replaced by the inhibition bimolecular rate constant, *k′_i_* = *k′*_2_/*k′_D_.*

However, in order to speed up the reaction, the encapsulated enzyme concentration, [*E*]_0_, has to be as high as possible. Therefore, under reaction conditions at high [*E*]_0_, even though the saturation kinetics is Michaelis–Menten-like, the standard Michaelian model does not apply. Indeed, for the derivation of Michaelis–Menten equations, it is assumed that [*E*]_0_ ≪ [*T*] and *k*_2_ ≪ *k*_−1_, so that under steady-state conditions, the depletion of the free substrate is negligible. On the contrary, in nRs, where [*E*]_0_ is high, a significant fraction of the substrate is bound to the enzyme as a reversible *E.T* complex. Thus, the concentration [*E.T*] must be included in the mass balance calculations for the enzyme and substrate:[*E*] = [*E*]_0_ − [*E.T*](17)
[*T*] = [*T*]_free_ = [*T*]_0_ − [*E.T*](18)

Therefore, velocity-dependent equations must include mass balances, taking into account substrate depletion (Equations (17) and (18)). Different formalisms have been developed to derive these equations, providing rate equations at high enzyme concentration [71,72,73,74,75]. The general rate equation is the one most widely used for the determination of catalytic parameters:(19)$$v=\frac{k{{}^{\prime }}_{\mathrm{cat}}}{2}\times \left\{\left({\left[S\right]}_{0}+{\left[E\right]}_{0}+{K^{\prime }}_{m}\right)-\sqrt{{\left({\left[S\right]}_{0}+{\left[E\right]}_{0}+{K^{\prime }}_{m}\right)}^{2}-4\;{\left[E\right]}_{0}\;{\left[S\right]}_{0}}\;\right\}$$

Equation (19) is valid at a high concentration of enzyme, when [*E*]_0_ is of the same order as [*T*]_0_. At saturation, a high [*T*]_0_ is:*v* = *k*’*_cat_* [*E*]_0_(20)
and at a low [*T*]_0_, Equation (5) becomes:(21)$$v=\frac{k{{}^{\prime }}_{\mathrm{cat}}}{\left(K{{}^{\prime }}_{m}+\left[E\right]\right)}\left[E\right]\left[T\right]$$

Experimental catalytic parameters can be determined from Equation (19). Data fitted to the Michaelis–Menten equation provides the apparent values of *K_m_* [71]:*K_m_* = *k′_m_* + [*E*]_0_(22)

The failure to account for the formation of products in the substrate mass balance ([*T*]_0_ = [*T*] + [*E.T*] + [*P*]) [73] has led to objections in the field to the use of Equation (19). However, when the velocity is determined using the initial rates, at which time the concentration of reaction products [*P*] is very small relative to [*T*]_0_, it does not affect the substrate mass balance value ([*T*]_0_ = [*T*] + [*E.T*]).

The high concentration of enzyme inside the nanoreactor, [*E*]*_nR_*, may have similar effects on enzyme-catalyzed reactions, such as viscosity, osmotic pressure, and macromolecular crowding. These three contributions interact, causing possible alterations in enzyme conformation, dynamics, and reaction kinetics. These effects are examined in the following sections.

#### 2.2.4. Viscosity Effects

Viscosity affects the rate at which reactions occur. The dependence of elementary kinetic constants on solvent viscosity provides information as to the extent to which a reaction is diffusion-controlled [76,77,78]. In nR, high viscosity is determined by the high concentration of the encapsulated enzyme.

The observed bimolecular rate constants (*k_II_*) for enzyme reactions (*k_cat_/K_m_* or *k*_2_*/K_D_*) are a combination of rate constants for two steps: the diffusion of toxicant molecules to the binding site, to form the productive reversible complex *E.T* (*k*_1_), the dissociation of this complex (*k*_−__1_), and the chemical step (*k*_2_) for reaction with the active site (Figure 1). Assuming the steady-state formation of the enzyme-inhibitor complex (*K_D_*) or enzyme-substrate complex (*K_m_*) [79], the bimolecular rate constant *k_II_* (= *k_cat_/K_m_* or = *k*_2_*/K_D_*) is:(23)$$k_{II}=\frac{k_{1}k_{2}}{\left(k_{-1}+k_{2}\right)}=\frac{k_{1}}{1+\frac{k_{-1}}{k_{2}}}$$
or:(23bis)$$\frac{k_{II}}{k_{1}}=\frac{1}{1+\frac{k_{-1}}{k_{2}}}$$

*k_II_* depends on the partition coefficient *P* = *k*_−__1_/*k*_2_. The dependence of *k_II_* on the relative viscosity leads (*η_rel_*) to the association rate constant *k*^0^_1_ and to the partition coefficient, *k*^0^_−__1_/*k*_2_. These constants can be determined by plotting the reciprocal of *k_II_* versus the relative viscosity [76] (Equation (24)):(24)$$\;\;\frac{1}{k_{II}}=\frac{1}{k_{1}^{0}}\eta_{rel}+\frac{k_{-1}^{0}}{k_{1}^{0}k_{2}}$$

The relative viscosity, *η_rel_* = *η*/*η*^0^ [79], is the ratio of the dynamic viscosity of the buffer containing viscosogen, *η*, to the viscosity of reaction buffer in the absence of viscosogen, *η*^0^.

The value of *P* allows us to determine whether the reaction is diffusion-limited or not. If *k*_−__1_ ≫ *k*_2_, there is a true equilibrium, the reaction is not limited by diffusion, and:(25)$$k_{II}=\frac{k_{1}k_{2}}{k_{-1}}=\frac{k_{2}}{K_{D}}$$

If *k*_−__1_ ≪ *k*_2_, then *P* = 0 and the reaction is diffusion-limited. In that case, the reaction with the enzyme would occur at a rate greater than the rate of the formation of complex *E.T*, and there would be a direct chemical modification of the enzyme’s active site, i.e., *k_i_* ≈ *k*_1_. If *k*_1_ ≈ *k*_2_, the reaction is partially diffusion-limited. However, this analysis can be complicated in the case of buried active centers, if the viscosogen cannot reach the active site. This situation also occurs when the viscosogen is the enzyme itself. In that case, the viscosogen also acts as an osmolyte.

A normalized plot for the dependence of *k_II_* on relative viscosity is described by Equation (26). This equation is the ratio of Equation (24), in which *η_rel_* = 1, i.e., *k_II_*^0^, divided by Equation (24), in which *η_rel_* > 1, i.e., *k_II_* [77]:(26)$$\frac{k_{II}^{0}}{k_{II}}=\left(\frac{P}{1+P}\right)+\left(\frac{1}{1+P}\right)\eta_{rel}$$

The slope of Equation (26) is *d*(*k*^0^*_i_*/*k_i_*)/*dη_rel_* = 1/(1 + *P*). For bimolecular reactions that are fully rate-limited by diffusion, *P* = 0 and the slope = 1. For reactions that are not diffusion-controlled, *P* is very large, making 1/(1 + *P*) very small, and the slope = 0. The slope of partially diffusion-controlled reactions ranges between 0 and 1. This approach describes the consequences of the reaction kinetics that arise from changing the viscosity of the medium. Effects due to viscosity should be the same regardless of the chemical nature of the viscosogen. If the slopes for plots of *k*^0^*_II_*/*k_II_ versus η_rel_* differ in the presence of different viscosogens, then perturbations of *k_II_* in addition to those due to the viscosity of the medium are occurring. If the slope is nonlinear, or if it exceeds 1, there is evidence of additional perturbations. Such perturbations include osmotic effects, viscosogen-induced inhibition of the reaction, or interactions of the viscosogen with the surface of the enzyme that alter the structure and/or dynamics of the enzyme. These latter effects are related to crowding.

#### 2.2.5. Osmotic Effects

Because of the permeability to water of the nanoreactor membranes, possible osmotic effects may arise in vivo if the concentration of encapsulated enzyme is very different from the concentration of osmolytes in the biological medium. Water molecules in both compartments will move until there is equality of thermodynamic potential on both sides of the nanoreactor membrane. For example, plasma proteins are the major determinant of blood osmotic pressure (*π*). Thus, taking the average protein concentration in plasma as 1 mM (i.e., 70 g/L with <M> = 70,000 Da), the injection of nanoreactors containing high concentrations of the encapsulated enzyme may induce osmotic effects if [*E*]*_nR_* is significantly different from this concentration. If [*E*]*_nR_* > 1 mM, water will enter the nanoreactor; if [*E*]*_nR_* < 1 mM, water will leave the nanoreactor. As a consequence, the volume of the nanoreactors will either swell or shrink (Δ*V* > 0 or < 0) (Figure 5). At the same time, the enzyme concentration and viscosity inside the nanoreactor will decrease or increase. This may affect the kinetics of encapsulated enzymes, either by changing the reaction order or by modifying the diffusion of reactants and products.

The effect of *π* on kinetic constants, *k*, of the enzyme reactions can be described using Equation (27) [80,81,82]:(27)$$\frac{\partial Lnk}{\partial \Pi }=-\frac{\Delta V^{‡}{}_{osm}}{RT}$$

In this equation, *π* = [*osmolyte*] × *R**T***, $\Delta V^{‡}{}_{osm}$ is the osmotic volume of activation (ml/mol); *R* is the gas constant (82.1 mL·atm·K^−1^·mol^−1^, where 1 atm = 1.013 bar = 0.1 MPa); and ***T*** is the absolute temperature (298 K). Thus, changes in *π* may induce the activation or inhibition of enzyme reactions, depending on the size and magnitude of $\Delta V^{‡}{}_{osm}$. Moreover, since osmolytes cause the removal of water molecules from the enzyme active sites, $\Delta V^{‡}{}_{osm}\;$ > 0 can be used to probe hydration changes in the enzyme (surface and active center) and the accompanying reactions. The maximum number of water molecules (*n_w_*), stripped off the hydration layer and out of the active site gorge, as a function of *π* is:(28)$$n_{w}=\frac{\Delta V^{‡}{}_{osm}}{v_{w}}$$

*v_w_*, the average molar volume of water in the active site, is equal to the average molecular volume of water multiplied by Avogadro’s number.

#### 2.2.6. Macromolecular Crowding

Crowding refers to the effects of high concentrations of macromolecules on chemical and biological processes. This has been extensively investigated in cellular physiology. Indeed, cells and biological fluids contain high concentrations of macromolecules, proteins, nucleic acids, and polysaccharides, i.e., 50–400 mg/mL [83]. Thus, a significant fraction of the cellular volume is occupied by macromolecules. As a consequence of crowding, compact enzyme conformations are favored and the accessible volume for substrates is reduced. The exclusion volume model of Minton [84] (Figure 6) satisfactorily describes the effect of crowding on enzyme structure and catalytic behavior [85].

Therefore, the kinetics of enzymes in highly concentrated and confined environments, like cytosol and nanoreactors, may be affected due to a decrease in the diffusion rate of substrates to active centers, altering the stability of *E.S* complexes and impairments in terms of conformational dynamics [86,87,88]. Anomalies in the diffusion of small molecules can be observed and described within the frame of fractal kinetics [89,90]. In addition, the stability of encapsulated enzymes is increased. Thus, high concentrations of encapsulated enzymes may affect the binding and catalytic parameters in nanoreactors.

### 2.3. Practical Implication of Second-Order Reactions in Nanoreactors

#### 2.3.1. General Considerations

Let us consider the detoxification reactions taking place under 0-, first- and second-order conditions (Figure 7A). Enzymologists are familiar with the classical formalism of classical enzyme kinetics, stating that [*E*] ≪ [*T*]. Under these conditions, the reaction order is progressively decreasing as a function of the substrate concentration, from 1, at concentrations much less than *K_m_*, to 0 at substrate saturation when the velocity is at maximum (*V_max_*). Indeed, at very low substrate concentrations, the enzyme reaction is first-order:*d*[*T*]/*dt* = *−k_I_*[*T*](29)
so that the substrate concentration decreases exponentially as a function of time:[*T*]_*t*_ = [*T*]_0_ *exp*(*−k_I_t*)(30)
with a constant half-time all along the process:*t*_1*/*2_*= Ln*2*/k_I_*(31)

Thus, at very low substrate concentrations, the first-order enzyme reaction rate (*k_I_*) is described by Equation (5) and the toxicant concentration decays exponentially according to Equation (6), with the half-time of the reaction (Equation (32)) being dimensionally similar to Equation (31):*t*_1*/*2_ = *Ln*2/(*k_cat_*.[*E*]/*K_m_*)(32)

As the substrate concentration is increasing, the reaction order is decreasing. At saturating substrate concentration, the reaction is zero-order:*d*[*T*]/*dt* = *k*_0_(33)

Thus, for zero-order reactions (*k*_0_), the substrate concentration decreases linearly:[*T*]_*t*_ = [*T*]_0_ − *k*_0_*t*(34)
with the half-time of the reaction,
*t*_1*/*2_ = [*T*]_0_/2*k*_0_.(35)

Thus, under order conditions of 0, *t*_1*/*2_ and [*T*] decrease linearly as the time that has passed increases.

Now, taking into account that the detoxification processes of toxic molecules, T, have to be achieved in very short times, the concentration of nR-encapsulated stoichiometric or catalytic bioscavengers, [*E*], has to be as high as possible. Under these conditions, the detoxification reactions take place under the second order (*k_II_*), i.e., [*E*] ≥ [*T*] and the toxicant concentration decays hyperbolically (Equation (37)):*d*[*T*]/*dt* = −*k_II_* [*T*]^2^(36)
(37)$${\left\lfloor T\right\rfloor }_{t}=\frac{{\left[T\right]}_{0}}{1+k_{II\;}{\left[T\right]}_{0}t}$$
with the half-time of reaction increasing concomitantly as [*T*] is decreasing,
*t*_1/2_ = 1/*k_II_* [*T*]_0_(38)

Unlike zero and first-order processes, in second-order reactions, the half-time increases as the concentration T decreases (Figure 7A). Therefore, effective second-order reactions of detoxification must lead to non-toxic concentrations of toxicant in a time period that is as short as possible. Thus, the first half-time of neutralization is short if the initial toxic concentration is high and if the bimolecular rate constant (*k_II_*) is high.

Another implication of second-order reactions is that [*T*] approaches zero more slowly than in first- and zero-order processes. Thus, in second-order kinetics, the bimolecular rate constants of the reacting enzymes (*k_II_* = *k_cat_*/*K_m_* for a catalytic bioscavenger or *k_i_* for a stoichiometric bioscavenger) have to be as high as possible to reach a non-toxic concentration [*T*] in the shortest time. In this condition, if *k_II_* > *k_I_*/*Ln*2.[*T*]_0_, then the initial decay of toxicant concentration is faster than in first-order process (Figure 7B). On the other hand, if *k_II_* is too slow, the second-order kinetics can be slower than in first-order processes (*k_II_* < *k_I_*/*Ln*2.[*T*]_0_) and the toxicant can persist in the bloodstream for a longer time (Figure 7B). One of the main challenges of the genetic engineering of detoxifying enzymes is to create enzymes displaying bimolecular rate constants that are as high as possible against potential toxicants.

#### 2.3.2. Stoichiometric and Pseudo-Catalytic Detoxification under Second-Order Processes

As discussed above, the stoichiometric neutralization reaction between E and a toxic substrate, T, is mole-to-mole, with *k_II_* = *k*_2_/*K_D_*, the bimolecular rate constant of the reaction (Figure 1a). The reaction leads to the formation of a covalent adduct, *E*-*T*’. This reaction is described by the equations for the irreversible inhibition of an enzyme under second-order conditions, e.g., active site titration. The enzyme is progressively inactivated: the concentrations of enzyme, [*E*], and toxicant, [*T*], decrease at the same time, while the concentration of the inactivated enzyme adduct ([*E-T’*] = *x*) at time *t* increases.
−*d*[*E*]/*dt* = −*d*[*T*]/*dt* = *d*[*x*]/*dt* = *k_II_* [*E*][*T*] = *k_II_* ([*E*]_0_ − *x*) ([*T*]_0_ − *x*)(39)

After integration, Equation (39) gives a linear relationship versus time:(40)$$Ln\frac{{\left[T\right]}_{0\;}–({\left[E\right]}_{0\;}–x)}{{\left[E\right]}_{0\;}–({\left[T\right]}_{0\;}–x)}=k_{II}\left({\left[E\right]}_{0\;}–{\left[T\right]}_{0\;}\right).t$$

*k_II_* and *t*_1/2_ can be determined by plotting Equation (40) data vs. time (cf. Figure 7). If [*E*]_0_ = [*T*]_0_, Equation (39) is simpler:*d*[*T*]/*dt* = *d*[*E*]/*dt* = *k_II_* ([*E*]_0_ − *x*)^2^(41)

Then:1/([*E*]_0_ − *x*)^2^ = *k_II_*.*t* + 1/[*E*]_0_(42)

Thus, for second-order reactions where *x* = [*T*]_0_/2 = [*E*]_0_/2, it follows that *t*_1*/*2_ = 1/*k_II_*[*T*]_0_ = 1/*k_II_*[*E*]_0_. Because t_1/2_ is inversely proportional to [*T*]_0_ or [*E*]_0_) (cf. Equations (35) and (42)), the higher the [*T*]_0_ and [*E*]_0_, the faster the detoxification process. This ideal case shows that detoxification under second-order conditions is more effective than detoxification under first-order conditions, if [*E*]_0_ and/or [*T*]_0_ are high (Figure 7B). However, if the bimolecular rate constant of the enzyme, *k_II_*, is too slow, the inactivation process may be slower than the first order-reaction (Figure 7B). Therefore, for effective stoichiometric or catalytic detoxification in enzymatic nanoreactors, both [*E*]_0_ and *k_II_* must be as high as possible.

In the particular case of the pseudo-catalytic inactivation of cholinesterase-based bioscavengers, the inactivated enzyme after the neutralization of the toxicant (E-T’) can be reactivated in a coupled reaction, leading to a pseudo-catalytic bioscavenger (Figure 1b and Figure 2). However, when a cycle of coupled consecutive reactions takes place inside the nanoreactors, additional constraints are imposed. Indeed, because toxicant molecules have to be destroyed in a very short time, both enzyme and reactivator concentrations must be high. Under these conditions, all reactions take place under second-order conditions within the confined nanoreactor volume. Thus, in the example of pseudo-catalytic bioscavengers against OPs, the encapsulated [*E*] has to be as high as the highest possible level [*OP*] present in the blood. In the most severe cases of OP poisoning, concentrations of the order of µM were observed. The concentration of the encapsulated reactivator, [*R*], which is much higher than both [*E*] and [*OP*] present inside nR, has to be in the order of mM to ensure the maximum reactivation rate and to balance possible leaks of the reactivator molecules out of the nanoreactor. One way to prevent leaks of the ChE reactivators from nR could be to covalently link the R molecules to the enzyme [91]. It must also be mentioned that the reaction products of reactivation, phosphyl oximes (OP’R), may irreversibly re-inhibit irreversibly the enzyme. However, phosphyl-oximes are unstable, and if they are leaking out of the nanoreactor, it is assumed that they are rapidly hydrolyzed in the bloodstream by endogenous phosphotriesterases (PTE), in particular, paraoxonase-1 (PON-1) [92].

#### 2.3.3. Catalytic Detoxification

If detoxification is performed by an enzyme using T as a real substrate, the detoxification process operates as for any enzyme reaction with a turnover where the active enzyme is recycled after each turnover (Figure 1c). For the detoxification of OPs, the most effective catalytic bioscavengers are PTEs (evolved bacterial PTEs and evolved mammalian PON-1) [93,94]. Under the second-order conditions of enzyme catalysis at high [*E*]_0_, the above-mentioned formalism applied to stoichiometric neutralization is operative, using the bimolecular rate constant of the catalytic reaction, *k_cat_/K_m_* = *k*_2_*/K_D_*. The recent achievement of an enzyme nanoreactor containing an evolved PTE for the detoxification of OPs showed that high concentrations of paraoxon (up to 5 mM) are completely hydrolyzed in less than 10 s by 1 mM of the enzyme. LD_50_-shift experiments in mice showed that the prophylactic and post-exposure intravenous administration of 1.6 nmoles of enzyme in nR provided animal protection/treatment against multiple LD_50_ of paraoxon without any other drugs (Pashirova et al., 2022, accepted for publication in ACS Applied Materials & Interfaces).

## 3. Conclusions

Therapeutic enzyme nanoreactors for the detoxification of endogenous compounds, e.g., for the destruction of endogenous toxicants, such as uric acid [4] or reactive oxygen species (ROS) produced in inflammatory processes [3], or drugs and exogenous toxicants [5] such as OPs (Pashirova et al., 2022, accepted for publication in ACS Applied Materials & Interfaces), have been successfully achieved.

Important physicochemical parameters determine the efficiency of enzymatic *nR*: (a) the stability and permeability of the envelope in terms of toxicants and reaction products; (b) the enzyme encapsulation efficiency and its concentration inside *nR* must be as high as possible; (c) the catalytic parameters of enzymes working in confined spaces are apparent where reaction kinetics are controlled by diffusion, viscosity, osmotic pressure, and possible macromolecular crowding. Moreover, the *nR*s intended to be administered in vivo must display operational stability, i.e., a long residence time, and be tolerated well. They must not induce an immune response or adverse iatrogenic effects. Their pharmacokinetics have to be optimized by selecting and eventually decorating the polymeric envelopes.

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
