# Peer review of "Kinetic Processes in Enzymatic Nanoreactors for In Vivo Detoxification"

_biomedicines, 2022, doi:10.3390/biomedicines10040784_

Round 1

Reviewer 1 Report

The review manuscript by Dr Masson presents principles of detoxification processes of toxicants by enzymes elucidating principles of kinetic mechanisms of enzyme reactions. The authors discuss the bases, practical consequences and applications of the detoxification in confined spaces such as nanoparticles or if the inactivating enzyme is freely circulating in the bloodstream, and the detoxification process takes place in plasma. Catalytic parameters are discussed in view of the Michaelis-Menten kinetics and effects on enzyme-catalysed reactions as viscosity, osmotic pressure and macromolecular crowding that cause possible alterations in enzyme conformation, dynamics and reaction kinetics. This manuscript on enzyme kinetics is very clearly written describing complicated reactions on a simple way, and therefore should attract a broad readership interested in encapsulation of biomolecules (e.g., antibodies, specific reactive proteins, target enzymes, chemically reactive traps such as cyclodextrines) as well as scholars interesting in enzyme kinetics.

Only a few minor corrections needed.
Page 1 – delete numbers for keyword
Page 1, abstract nannoreactor should read nanoreactor
Page 2, line 2 - Add full stop after ref2
Page 5, line 2 from bottom – add nanoreactor for abbreviation nR 
Page 6. edit the sentence “Moreover, a high concentration of enzyme(s) is present this small volume.“ 
Page 6, para 2.2.1. after ref 23-27 add comma
Page 6. Sentence “Increase in enzyme loading into nanoreactor [30] developed this issue.” should not be one paragraph.
Page 7. Add numeration if “Improved permeability to reactants and reaction products” is title, or correct as a sentence.
Page 8. move „and NnR the nanoreactors concentration (particle number concentrations/mL)“ to next paragraph
Figure 3 is not referred in text.
Page 8, para 3 – replace „and“ „NnR (particle number concentrations) and it is possible“ with comma
Page 8. After ref. 47 add full stop
Page 17. Delete symbol after ref. 3

Author Response

P1: keywords numbers were deleted

P1: nanoreactor in abstract was corrected

P2: full stop after ref 2

P5: nanoreactor (nR) corrected

P6: the sentense « Moreover, high concentrations of enzyme(s) are present in this small volume. » was edited

P6: comma vas added after ref 23-27

P6: sentense was reintroduced in the text before the following paragraph.

P7: sentense corrected

P8: sentense was moved to next paragraph and Figure 3 was mentioned in the  paragraph located just above this figure

P8: full stop after ref 47.

P17: comma deleted after ref 3

Reviewer 2 Report

Starting at paragraph 2.1.1., the description of the methods becomes confusing. The different kinetic constants (k1, k-1, k2, k3) do not seem to refer to any shown kinetic scheme, and it is quite hard to move on with the reading. Authors should give more attention to this essential point.

Author Response

We agree, the beginning of paragraph 2.1.1 was confusing. To clarify the point addressed by Reviewer 2, we wrote an additional mechanism in Scheme 1 (Scheme 1d), corresponding to the situation described by equations containing constants k1, k-1, k2and k3. While Scheme 1c describes the simplest catalytic detoxification process, Scheme 1d describes two-step catalytic detoxification process such as mechanisms of hydrolases (e.g. hydrolysis of toxic compounds by cholinesterases and carboxylesterases). Corresponding equations for Scheme 1d were inserted in the text (Eq. 3) after those for Scheme 1c (Eq. 3).

Reviewer 3 Report

The manuscript by Masson and coworkers details enzymatic nanoreactors. Specifically, their manuscript focuses on the inactivation of organophosphorus compounds by bio-scavengers’ enzymes. The manuscript tries to encompass all the theoretical considerations to validate the concept. The organization and details are good and concise. Since the article talks about detoxifying some compounds. It will be better to include a table with known enzymatic nanoreactors and one more table or scheme with the chemical structures of the potential compounds to be neutralized in order of reactivity.

Author Response

On P3 (top), we refer to several articles (ref 3-7) about enzyme nanoreactors for detoxification of endogenous and exogenous toxicants.

New figure 1 shows structures of 3 typical organophosphorous compounds, paraoxon, echothiophate and sarin. On the next page, lines 12-14, the nature of X, the leaving group of these compounds is explained.